

# Impact of typhoons on particulate and dissolved $^{137}$Cs activities in seawater off the Fukushima Prefecture: results from the SOSO 5 Rivers cruise (October 2014)

Michio Aoyama [1,2], Sabine Charmasson [3], Yasunori Hamajima [4], Celine Duffa [3]

[1] Centre for Research in Environmental Dynamics, Univ. of Tsukuba, 1-1-1 Ten-noudai, Tsukuba 305-8572, Japan

[2] Institute of Environmental Radioactivity, Fukushima Univ. Kanayagawa 1, Fukushima 960-1296, Japan

[3] Institute de Radioprotection et de Sûreté nucléaire (IRSN), PSE-ENV, LRTA, BP3, 13115 Saint Paul lez Durance, France

[4] The Institute of Nature and Environmental Technology, Kanazawa University, Wake, Nomi, Ishikawa 923-1224, Japan

*Correspondence to*: Michio AOYAMA (michio.aoyama@ied.tsukuba.ac.jp)

**Abstract.** Cruise SoSo 5 Rivers took place during October 2014 off the coast of Fukushima Prefecture shortly after the passage of two typhoons. Detection of dissolved $^{134}$Cs and $^{137}$Cs in all samples reflected contamination caused by accidental releases of radiocaesium from the Fukushima Dai-ichi Nuclear power plant (FNPP1) accident. The dissolved activities were generally higher at coastal sites and decreased with distance from shore, and they were higher in the surface than in the bottom water. The tendency of $^{137}$Cs activities to decrease with distance from the coast reflected mixing of coastal water and open-ocean

water of which $^{137}$Cs activity concentration was ~1.5 Bq m$^{-3}$. At stations very close to the coast, we observed high particulate $^{137}$Cs activity concentration that exceeded dissolved $^{137}$Cs activity concentration. $^{137}$Cs activities were generally 1–2 orders of magnitudes lower in organic particles than in dissolved form, and the ratios of $^{137}$Cs activity concentration in organic particles to $^{137}$Cs activity concentration in dissolved form ranged from 0.01 ± 0.00 to 0.12 ± 0.01. The ratio of $^{137}$Cs to $^{134}$Cs activity concentrations in organic particles did not change with distance from shore or with $^{137}$Cs activity concentration and generally

remained around 1, even in samples collected far from the coast. This pattern indicated that the organic particles had come from rivers or a source very close to the coast. The $^{137}$Cs/$^{134}$Cs activity ratio in dissolved form north of FNPP1 region was estimated to be 1.074 ± 0.015, a ratio that is in good agreement with the $^{137}$Cs/$^{134}$Cs activity ratio in the core of Unit 1 of the FNPP1 while the $^{137}$Cs/$^{134}$Cs activity ratio at Tomioka port which located south of FNPP1 was 0.998 ± 0.017. Therefore we can conclude the source of radiocaesium in seawater in the coastal region north of FNPP1 was deposited radiocaesium released

from the core of Unit 1 of FNPP1, while the source of radiocaesium observed in the coastal region south of FNPP1 was a mixture of deposited radiocaesium released from the core of Unit 2 and the core of Unit 1 of FNPP1. During September–October of each year, the typhoon season in Japan, the $^{137}$Cs activity concentration generally increased at Ukedo port, Tomioka port, FNPP1, and Iwasawa beach, and showed a good relationship with the 7-day modified antecedent precipitation index (API) while there is less correlation between the modified API and $^{137}$Cs activity concentration near the outlet of canal from

unit 5 and 6 of FNPP1 to the sea.



## 1 Introduction

The Fukushima Dai-ichi Nuclear Power Plant (hereafter FNPP1) accident, which occurred in northeastern Japan on 11 March 2011, resulted in releases of large amounts of various radionuclides, especially $^{137}$Cs and $^{134}$Cs, which have half-lives of 30.2

y and 2.06 y, respectively. Estimates tend to converge on 15–20 PBq for the combined inputs of $^{137}$Cs from atmospheric fallout and direct discharge to the North Pacific (Aoyama et al., 2019;Aoyama et al., 2016). According to Nishihara et al. (Nishihara et al., 2012), the $^{137}$Cs to $^{134}$Cs activity concentration ratios in reactor cores 1, 2, and 3 were 1.06, 0.92, and 0.96, respectively, at the time of the accident. In the coastal region, however, the $^{137}$Cs to $^{134}$Cs activity concentration ratio in dissolved radiocaesium was reported to be uniform and very close to 1 (Buesseler et al., 2012;Buesseler et al., 2011), probably because

of inaccuracies in the measurements made during monitoring by the Japanese Government and Tokyo Electric Power Company Holdings, hereafter TEPCO. Small but significant differences have already been reported in the $^{137}$Cs to $^{134}$Cs activity concentration ratios of the environmental samples collected on land. Those differences depended on whether the radiocaesium came from the unit 1, 2, or 3 reactor core. Miura et al. (Miura et al., 2020) have reported two types of caesium-bearing microparticles emitted from the FNPP1 accident that were separated from road dust and non-woven fabric cloth. Type-A

particles, which were spherical and ~0.1–10 μm in diameter, contained ~$10^{-2}$ to $10^{2}$ Bq of $^{137}$Cs radioactivity concentration. Type-B particles, which had various shapes and were 50–400 μm in diameter, contained $10^{1}$–$10^{4}$ Bq of $^{137}$Cs radioactivity concentration. The $^{137}$Cs to $^{134}$Cs activity concentration ratios in Type A and Type B particles were in excellent agreement with the corresponding ratios in cores 2 and 3 in the former case and in core 1 in the latter case. The evidence therefore indicated that the type A particles came from Units 2 and 3, and the Type B particles came from Unit 1. There has not been a

similar study of the dissolved and particulate ratios of $^{137}$Cs to $^{134}$Cs activity concentration in seawater.

Although these major releases from the FNPP1 accident occurred in March and April 2011 (Tsumune et al., 2012;Tsumune et al., 2013), small amounts of radiocaesium have continued to be released, and the rate of release of $^{137}$Cs was estimated to be 10 GBq day$^{-1}$ or 3.6 TBq year$^{-1}$ in 2014 (Tsumune et al., 2020). Also, riverine inputs due to runoff from contaminated watersheds, though of lesser importance, are expected to continue for a long time (Adhiraga Pratama et al., 2015). Movement

of primarily particulate radiocaesium in fresh water into the Pacific Ocean appears to have occurred during extreme weather events because of the high affinity of Cs for particles. The result was the transfer of up to 10–12 TBq of radiocaesium from the land to the ocean during the first year after the accident (Evrard et al., 2015). (Nagao et al., 2014) found that about 50% of the radiocaesium in the Niida River was in dissolved form in the first few months after the accident, whereas after September 2011 more than 70% of the radiocaesium was associated with particles, and that percentage was even higher after storms.

Recently, inputs of particulate and dissolved radiocaesium by several rivers in the Maeda Basin following heavy rain events including typhoons were estimated (Sakuma et al., 2019). They have developed a simple model to estimate the $^{137}$Cs discharge from catchments. They used this model to estimate the discharge of $^{137}$Cs and the ratio of discharged $^{137}$Cs to the inventory of $^{137}$Cs deposited in the catchments of the Abukuma River and 13 other rivers in the Fukushima coastal region from the time immediately after the Fukushima accident until December 2017. The discharge of $^{137}$Cs and the ratio of discharged $^{137}$Cs to

the inventory of $^{137}$Cs during the initial six months after the accident were estimated to be 18 TBq (3.1%) for the Abukuma





River and 11 TBq (0.79%) for the 13 other rivers. These $^{137}$Cs discharge ratios were 1–2 orders of magnitude higher than those observed after June 2011 in previous studies (Ueda et al., 2013;Tsuji et al., 2016;Iwagami et al., 2017). The impact on the ocean from the initial discharge of $^{137}$Cs through the rivers was limited because the discharge of 29 TBq (18 TBq + 11 TBq) of $^{137}$Cs from the Abukuma River and the 13 other rivers in the Fukushima coastal region was two orders of magnitude smaller

than the direct release into the ocean of $3.5 \pm 0.7$ PBq from the FNPP1 (Tsumune et al., 2012) and 7.6 PBq from atmospheric deposition (Kobayashi et al., 2013). During the period from October 2012 to December 2017, the total discharge from 13 rivers in the Maeda basin has been estimated to have been 12 TBq (23 TBq − 11 TBq) (Sakuma et al., 2019). Direct discharge from the FNPP1 site decreased dramatically during that time, and the direct discharge of $^{137}$Cs to the ocean was estimated to be only 0.73–1.0 TBq year$^{-1}$ during 2016–2018 (Aoyama et al., 2020d). This pattern indicates that discharges from the 13 rivers in the

Maeda basin and direct discharges might have become similar in magnitude recently. Among the 13 rivers in the Fukushima coastal region, the Ukedo River was the major contributor of $^{137}$Cs discharged to the ocean during the study periods of 11 March 2011 to 27 September 2012 and 11 March 2011 to 31 December 2017 because the total inventory of $^{137}$Cs in the catchment of the Ukedo River derived from atmospheric fallout was the largest component (536 PBq) of the total amount of $^{137}$Cs in the catchments of the 13 rivers in the Fukushima coastal region, 1282 PBq (Sakuma et al., 2019). The results of

Sakuma et al. (2019) showed that the particulate $^{137}$Cs discharge from the Abukuma River was approximately 1–2 orders of magnitude larger than the dissolved $^{137}$Cs discharge under both base flow and storm flow conditions. The particulate $^{137}$Cs discharge from the Maeda River was close to the dissolved $^{137}$Cs discharge under base-flow conditions, but the discharge of particulate $^{137}$Cs was approximately 1–3 orders of magnitude larger than the discharge of dissolved $^{137}$Cs under storm flow conditions. We therefore expected that the discharge of particulate $^{137}$Cs might account for most of the discharge of $^{137}$Cs from

rivers in our study region after heavy rains associated with typhoons.

The goal of this study, which was conducted in October 2014, was to analyse coastal waters close to small coastal rivers that flowed through highly contaminated watersheds close to the damaged FNNP1 after the passage of two typhoons. We examined dissolved $^{137}$Cs activities and all forms of particulate $^{137}$Cs activity concentration in seawater, as well as the ratios of $^{137}$Cs to $^{134}$Cs activities. We also analyzed the long-term trend of radiocaesium data in surface water at Tomioka together with

precipitation records and monitoring data obtained by TEPCO to understand riverine fluxes in the coastal regions of Fukushima.

## 2. Material and methods

In October 2014, the cruise SOSO 5 Rivers took place off the coast of Fukushima prefecture. The sampling targeted the areas off the mouths of five rivers (i.e., the Mano, Nitta, Ota, Odaka, and Ukedo rivers) located north of the FNPP1, the watersheds

of which were highly contaminated by fallout from the FNPP1 accident. Seawater was sampled at the surface and from 1 m above the sea bottom at 5 stations along 5 radial transects from the mouths of each of these five rivers as total 25 stations as shown black dots in Fig. 1. We also collect surface seawater at 12 locations, S1 to S12, along the coast from north of the mouth



of the Mano River at station S1 to south of the mouth of the Ukedo River at the stations S11 and S12 where the water depths were 8–10 m as shown  red solid circles in Fig. 1. In addition, one water sample was collected from four of the five estuaries

(the Ukedo estuary was not accessible at the time of sampling, locations are not shown in Fig.1). All samples were filtered through 0.45-μm filters.

Before the cruise, typhoon #18 (Phanfone) caused heavy rain in the area of interest on 6 October 2014. We collected samples along the Mano River transect on 9 October and along the Ota River transect on 10 October. Because typhoon #19 (Vongfong) made landfall on 13 to 14 October, sampling was postponed until 17 October for the Ukedo River transect, 18 October for the

Odaka River transect, and 19 October for the Nitta River transect. Sampling at all stations S1–S12 was carried out between 16 and 19 October.

An improved ammonium phosphomolybdate (AMP) procedure (Aoyama and Hirose, 2008) was used to extract radiocaesium from the seawater samples as dissolved form which were obtained by filtration through 0.45-μm filters. With this procedure, the weight yields of the AMP/Cs compound as well as the radiochemical yields of radiocaesium generally exceeded 99% for

2-liter samples. The activities of AMP/Cs compounds were measured at the Ogoya Underground Facility of the Low Level Radioactivity Laboratory of Kanazawa University using high-efficiency, well-type, ultra-low-background Ge-detectors (Lutter et al., 2015;Aoyama et al., 2009;Hamajima and Komura, 2004). Organic form of radiocaesium of the samples were obtained by disillusion of organic portion on the filter using concentrated nitric acid and concentrated hydrogen peroxide at all stations, then filled and dried up in a Teflon tube. The activities of organic portion of radiocaesium in Teflon tube were also measured

at the Ogoya Underground Facility of the Low Level Radioactivity Laboratory At some stations, we were able to recover a sufficient amount of suspended matter (by filtration through 0.45-μm filters) to measure radiocaesium activities by gamma spectrometry at the Modane underground laboratory in France using ultra-low-level, well-type, high-purity Ge γ-detectors (Canberra Industries). The counting time ranged between 240,000 and 320,000 s. Because these samples were collected separately from other samples and the suspended matter was heterogeneous, the mass concentrations of samples for all particles and for organic particles were not the same. However, the $^{137}$Cs and $^{134}$Cs activities in the organic particles and all particles

did not depend on the mass concentrations. We have therefore reported the activities of radiocaesium in both kinds of particles in this article.

At Tomioka port (open circle in Fig. 1), we measured the dissolved radiocaesium activity concentration in all samples and the particulate radiocaesium activity concentration for selected samples during the period from June 2014 to April 2019 using the

methodology described above.

## 3. Results

Activity concentrations of $^{134}$Cs and $^{137}$Cs in dissolved form, all particles, and organic particles collected during the SoSo 5 Rivers cruise activities used in this study are in a published dataset entitled "Dataset of 134Cs and 137Cs activity concentration



concentrations in dissolved for, all particles and organic form of particles obtained by SoSo 5 rivers cruise in 2014" as doi:
10.34355/CRiED.U.TSUKUBA.00030 (Aoyama et al., 2020a).

Activity concentrations of $^{134}$Cs and $^{137}$Cs in dissolved form  and all particles collected at 12 stations off Tomioka on 28 August 2014 used in this study are in a published dataset entitled "Dataset of 134Cs and 137Cs activity concentrations in dissolved form and, all particles at Tomioka, Fukushima in August 2014" as doi: 10.34355/CRiED.U.TSUKUBA.00031 (Aoyama et al., 2020b).

Activity concentrations of $^{134}$Cs and $^{137}$Cs in dissolved form collected at Tomioka port during the period from 10 June 2014 to 24 April 2019 used in this study are in a published dataset entitled "Dataset of time series of radiocaesium activity concentrations at Tomioka, Fukushima during the period from 10 June 2014 to 24 April 2019" as doi: 10.34355/CRiED.U.TSUKUBA.00032 (Aoyama et al., 2020c).

We also used TEPCO time series monitoring data of $^{137}$Cs activity concentration in surface water at Ukedo port, near the outlet
of canal from unit 5 and 6 of FNPP1 to the sea, hereafter 56N of FNPP1, and Iwasawa beach (open circles in Fig. 1) and at the Fukushima Dai-ni Nuclear Power Plant, hereafter FNPP2 (solid square in Fig. 1). TEPCO time series data used in this study are available at https://emdb.jaea.go.jp/emdb/portals/1060113000/.

## 3.1 Dissolved radiocaesium

The detection of both $^{134}$Cs and $^{137}$Cs in all samples confirmed contamination from the FNPP1 accident. Generally, the dissolved concentrations were higher at coastal sites and decreased with distance from the coast (Fig. 2), and they were higher in the surface layer compared to the bottom layer (Fig. 3).

At the stations very close to the coast within 1 km from the coast, $^{137}$Cs activity concentration ranged from $10.6 \pm 0.6$ Bq m$^{-3}$ at the bottom of Niida 1 station to $62.5 \pm 3.5$ Bq m$^{-3}$ at the surface of station S12 as shown in Fig.3 and in a dataset doi:
10.34355/CRiED.U.TSUKUBA.00030 (Aoyama et al., 2020a). They then decreased to less than 10 Bq m$^{-3}$ at the stations about 2 km from the coast for bottom samples and at the stations about 4 km from the coast for surface samples along each radial transect (Fig. 3). In general, the lowest $^{137}$Cs activity concentration was observed at the bottom at the stations furthest from the coast (Fig. 3). The lowest $^{137}$Cs activity concentration was therefore observed at the bottom at the Nitta 5 station ($2.0 \pm 0.1$ Bq m$^{-3}$), whereas the highest $^{137}$Cs activity concentration was observed at the surface at station S12 ($62.5 \pm 3.5$ Bq m$^{-3}$)
as shown in the dataset  doi: 10.34355/CRiED.U.TSUKUBA.00030 (Aoyama et al., 2020a).

The decreases of $^{137}$Cs activities in surface water with distance from shore were similar along four of the radial transects (the Mano, Niida, Odata, and Ota transects). The activity concentration at a distance of 1 km decreased to 90–96% of that at the first station along each transect and 36–68% of the corresponding activity concentration at a distance of 10 km. The rate of decrease was higher along the Ukedo transect: 85% at 1 km and 19% at 10 km. These differences might reflect differences in
the magnitudes of the river fluxes associated with the heavy rains caused by the typhoons that passed over the study area and the initial $^{137}$Cs activities in the river water.



The entire dataset showed clearly higher activities at all the locations nearest the coast, regardless of salinity. In general, $^{137}$Cs activities tended to decrease with distance from the coast. The $^{137}$Cs activities also tended to decrease with increasing salinity (Fig. 4) because of mixing of coastal water and open-ocean water in which the $^{137}$Cs activity concentration was ~1.5 Bq m$^{-3}$

(see section 4.2).

### 3.2 Particle-associated radiocaesium

The $^{137}$Cs activity concentrations in all particles observed in this study ranged from $2.25 \pm 0.33$ Bq m$^{-3}$ in surface water at Odaka 1 to $704 \pm 88$ Bq m$^{-3}$ in surface water at Ukedo 1. At the stations very close to the coast, Ukedo 1, S4, S10, S11, and S12, we observed $^{137}$Cs activity concentration in all particles exceeded the dissolved $^{137}$Cs activity concentration (Fig. 5). The

170 ratio of particulate $^{137}$Cs activity concentration to the sum of particulate and dissolved $^{137}$Cs activity concentration ranged from $0.13 \pm 0.02$ to $0.96 \pm 0.13$ at all stations. Ratios at the lower end of this range are comparable to the ratios observed at Tomioka in August 2014. Those ratios ranged from $0.06 \pm 0.01$ to $0.18 \pm 0.01$ during a time of not much rain (Aoyama et al., 2020b). Particulate $^{137}$Cs activities in organic particles were generally one or two orders of magnitude lower than dissolved $^{137}$Cs activities, and the ratio of $^{137}$Cs activity concentration in organic particles to dissolved $^{137}$Cs activity concentration ranged from

175 $0.01 \pm 0.00$ to $0.12 \pm 0.01$, except at a few stations. The ratios at those exceptional stations were $1.18 \pm 0.09$ at the bottom at Odaka 4, $0.15 \pm 0.01$ at the surface of Ota 3, and $0.32 \pm 0.03$ at the surface of station S4.

### 3.3 Activity ratios of $^{137}$Cs to $^{134}$Cs in dissolved form, all particles, and organic particles

The ratio of dissolved $^{137}$Cs to dissolved $^{134}$Cs activity concentration changed because of mixing with open-ocean water of

180 which $^{137}$Cs activity concentration is ~1.5 Bq m$^{-3}$ and $^{134}$Cs activity concentration can be assumed zero due to relatively short half-life (Figs. 6 and 7). Along a radial transect, the ratio of dissolved $^{137}$Cs to dissolved $^{134}$Cs activity concentration tended to increase with distance from the coast (Fig. 6). Mixing with open-ocean water tended to increase the ratio of $^{137}$Cs to $^{134}$Cs activity concentration because the open-ocean seawater contained virtually no $^{134}$Cs. It is very clear from Fig. 7 that the ratio of dissolved $^{137}$Cs to dissolved $^{134}$Cs activity concentration increased as the dissolved $^{137}$Cs activity concentration decreased

because of mixing between open-ocean seawater and freshwater (see section 4.2). In contrast, the ratio of $^{137}$Cs to $^{134}$Cs activity concentration in organic particles did not change with distance and remained at around 1 along 5 radial transects of SoSo 5 river cruises. This observation indicates that the source of the organic particles was very close to the coast, or that the particles were present in the rivers, indeed Naulier et. al. (Naulier et al., 2017) showed that organic particulate matter from contaminated watersheds could be a significant radiocaesium carrier towards the sea.

## 4. Discussion

### 4.1 An estimation of initial activity concentration of freshwater from land

There was no clear relationship between salinity and dissolved radiocaesium activity concentration (Fig. 4), except off the mouth of the Ukedo River where dissolved radiocaesium activity concentrations clearly decreased with increasing salinity



(Fig. 8). This relationship could be the signature of riverine inputs to the coastal areas because the Ukedo River flows through a highly contaminated watershed. However, several authors have emphasized that rivers discharge small amounts of dissolved radiocaesium (Sakuma et al., 2019;Nagao et al., 2014). As mentioned in the Introduction, during the period of our sampling, typhoons #18 (Phanfone) and #19 (Vongfong) made landfall on Japan on 6 and 13 October, respectively. These typhoons were associated with heavy rainfall and runoff that lowered the salinity of coastal waters and increased the fluxes of radionuclides

from the FNPP1 accident. Because the Ukedo River is close to the FNPP1, this relationship between salinity and dissolved radiocaesium activity concentration off the mouth of the Ukedo River can be linked to runoff of contaminated water because the highest $^{137}$Cs activity concentration during the typhoons was observed in the 56N canal of the FNPP1 (Fig. 9)).

A simple mixing model that explained the $^{137}$Cs data along the Ukedo radial transect on 17 October 2014 showed that the $^{137}$Cs activity concentration at a salinity of zero (i.e., the $^{137}$Cs activity concentration in the Ukedo River) should be around 260 Bq

m$^{-3}$ (Fig. 8). This $^{137}$Cs activity concentration is in good agreement with the activity concentration of 230 Bq m$^{-3}$ reported in the TEPCO monitoring data at the Ukedo port on 16 October 2014 (Fig. 9). In contrast, off the mouths of the other four rivers, a negative correlation or weak relationship (data not shown) was found between salinity and radiocaesium activity concentration, though the range of salinity was relatively small. In the case of the Ota River, where the correlation was positive, the most-saline water was found near the coast, and lower salinity water was found further from the river mouth (figure not

shown; data are in Aoyama et al., 2020a). This pattern might reflect complex physical processes associated with southward advection and a small amount of eddy mixing along the coast.

### 4.2 Source term estimation of observed radiocaesium based on $^{137}$Cs/$^{134}$Cs activity ratio in seawater

We carried out standardized major axis regression which accounts for the uncertainty in both x and y by minimizing the errors

in both directions on decay-corrected (to 11 March 2011) $^{134}$Cs and $^{137}$Cs activity concentrations in the samples collected by the SoSo 5 rivers project (Aoyama et al., 2020a) and the time series of observations at Tomioka (Aoyama et al., 2020c). Then we obtained the $^{137}$Cs/$^{134}$Cs activity ratio at the time of the accident as a slope of standardized major axis regression line and the pre-Fukushima $^{137}$Cs activity concentration due to global fallout from atmospheric weapons tests as an intercept of regression line. Results of a standardized major axis regression are shown in Table 1 and Figures 10 and 11. The $^{137}$Cs/$^{134}$Cs

activity ratios in the cores of the FNPP1 at the time of the accident has been estimated (Nishihara et al., 2012) and are also shown in Table 1.

The relationship between the decay-corrected $^{137}$Cs and $^{134}$Cs activities at the time of the accident (Fig. 10) showed that the pre-Fukushima $^{137}$Cs activities due to global fallout from atmospheric weapons tests were 1.3 ± 0.2 Bq m$^{-3}$, in agreement with the pre-Fukushima $^{137}$Cs activity concentration reported (Aoyama et al., 2008). Tsuruta et al., (Tsuruta et al., 2014) have shown

that the first plume from Unit 1 of the FNPP1 contaminated the catchment area of the Ukedo River. The $^{137}$Cs/$^{134}$Cs activity ratio in dissolved form in the SoSo project samples associated with the FNPP1 accident was estimated to be 1.074 ± 0.015. This $^{137}$Cs/$^{134}$Cs activity ratio is in good agreement with the $^{137}$Cs/$^{134}$Cs activity ratio of 1.06 (+-10%) of the radiocaesium in the core of Unit 1 of the FNPP1 at the time of the accident (Nishihara et al., 2012).



The pre-Fukushima [137]Cs activity concentration originated from atmospheric weapons tests in the seawater samples collected

at Tomioka port was estimated to be $1.2 \pm 0.7$ Bq m$^{-3}$ (Fig. 11), again in good agreement with the activity concentration in the SoSo project samples and previous reported pre-Fukushima [137]Cs activity concentration (Aoyama et al., 2008). The [137]Cs/[134]Cs activity ratio in the dissolved radiocaesium in surface water at Tomioka port was $0.998 \pm 0.017$. The Tomioka port located 10 km south of the FNPP1 (Fig. 1) and also locates at the mouth of the Tomioka river. In the Tomioka River catchment, the source of radiocaesium might be the core of Unit 2 of FNPP1 (Nakajima et al., 2017), then the [137]Cs/[134]Cs activity concentration ratio

might be 0.92 (+-10%) (Nishihara et al., 2012). The [137]Cs/[134]Cs activity ratio of $0.998 \pm 0.017$ observed at Tomioka port was, however, slightly higher rather than that in the core of Unit 2 of the FNPP1. Possible explanation of this finding are that the radiocaesium in the coastal seawater at the Tomioka port might be a mixture of radiocaesium from the core of the Unit 1 (ratio = 1.06) and the Unit 2 (ratio = 0.92) of the FNPP1 because the flow direction of the coastal current is primarily southward in this region.

Therefore we can conclude the source of radiocaesium in seawater in the coastal region north of FNPP1 was deposited radiocaesium released from the core of Unit 1 of FNPP1, while the source of radiocaesium observed in the coastal region south of FNPP1 was a mixture of deposited radiocaesium released from the core of Unit 2 and the core of Unit 1 of FNPP1.

### 4.3 Analysis of relationship between [137]Cs activity concentration in surface water and the antecedent precipitation index

During our concentrated study period in October 2014, we observed increase of [137]Cs activity concentration in seawater which might reflect increased riverine flux of [137]Cs due to heavy rain as discussed in the previous sections. We therefore investigated the relationship between [137]Cs activity concentration in seawater observed at several stations along the Fukushima coast and precipitation observed by Automated Meteorological Data Acquisition System located at Tomioka Town (Fig. 1). We adapted the antecedent precipitation index, hereafter API, which is a weighted summation of daily amounts of precipitation and is used

as an index of soil moisture (https://glossary.ametsoc.org/wiki/Antecedent_precipitation_index, accessed on 6 October 2020) as an index of the fluvial flux of radiocaesium in this study. The API is usually calculated with equation (1) .

$$API = P_0 + k^1 P_1 + k^2 P_2 + , , , + k^n P_n \tag{1}$$

where $P_0$ is total amount of precipitation on the day of radiocaesium measurement, $P_n$ is total amount of precipitation on day n before radiocaesium measurement, k is constant that depends on region and season and n is period of calculation.

In this study, we did not have information about k and n. We set k equal to 1 and let n vary between 1 and 9. We also did not include $P_0$ because sampling was generally conducted in the morning, and there was a time lag between a rainfall event in the

catchment of each river and the time the associated runoff reached the coast. We therefore modified the API in this study. Because of the tsunami and radioactive contamination from the FNPP1, the precipitation-observation system was not available until April 2014, the modified API could be calculated only after April 2014.



We calculated the correlation coefficients between the 1–9-day modified APIs and the $^{137}$Cs activity concentration in surface water at Ukedo port, the 56N of the FNPP1, Tomioka port, and Iwasawa beach. In Figs. 12a to 12e, we also show examples

of relationships between modified API and the $^{137}$Cs activity concentration in surface water at Tomioka port for 1 day, 3 days, 5 days, 7 days and 9 days APIs. As shown in Figs. 12 to 16 and Fig. 17, the correlation became better with increase of days of integration from 1 day to 5 days to calculate API. The correlation coefficient reached a maximum for 5–7-days modified APIs at Tomioka port, Ukedo port, and Iwasawa beach (Fig. 17) and since most of the seawater sampling was conducted for ~7 days interval, we chose the 7-day modified API as an appropriate API index in this study hereafter.

To show general figure on the relationship between 7 days modified API and the $^{137}$Cs activity concentrations, a Hovmöller diagram of the $^{137}$Cs activity concentrations at Ukedo port, the 56N of the FNPP1, Tomioka port, at FNPP2 port and Iwasawa beach is presented with the 7-day modified API with a reversed ordinate in this Figure (Fig. 18). It is clear that in September– October of each year, the typhoon season in Japan, the 7-day modified API exceeded ~150 mm and then the $^{137}$Cs activity concentration increased at Ukedo port, Tomioka port, FNPP2, and Iwasawa beach, but the relationship with heavy rainfall was

weak in the 56N of the FNPP1. The reason for the lower correlation coefficient between $^{137}$Cs activity concentration at the 56N of the FNPP1 and the 7-day modified API might reflect occurring extreme increases of $^{137}$Cs activity concentration without heavy rain events within the FNPP1 site as show in Fig. 19.

## 5. Conclusions

The detection of dissolved $^{134}$Cs and $^{137}$Cs activity concentrations in all samples demonstrated contamination from the FNPP1 accident. The dissolved $^{137}$Cs activity concentrations were generally higher at coastal sites and decreased with distance from the coast, and they were higher in the surface layer compared to the bottom layer. The decreases of the rates of $^{137}$Cs activity concentration with distance from shore in surface water were similar along radial transects at Mano, Niida, Odata, and Ota. The decrease of $^{137}$Cs activity concentration with increasing salinity reflected mixing of coastal water with open-ocean water,

the $^{137}$Cs activity concentration of which was only ~1.5 Bq m$^{-3}$.

$^{137}$Cs activity concentration in all particles observed in this study ranged from $2.25 \pm 0.33$ Bq m$^{-3}$ at the surface at Odaka 1 to $704 \pm 88$ Bq m$^{-3}$ at the surface at Ukedo 1. At the stations very close to the coast, Ukedo 1, S4, S10, S11, and S12, we observed relatively high $^{137}$Cs activities in all particles, and those activities exceeded the dissolved $^{137}$Cs activities. The ratio of $^{137}$Cs activity concentration in all particles to the sum of dissolved and particulate $^{137}$Cs activity concentration ranged from $0.13 \pm$

$0.02$ to $0.96 \pm 0.13$ at all stations and the lower range of this ratio was consistent with the analogous ratios at Tomioka port in August 2014 when there was no heavy rainfall.

$^{137}$Cs activities were generally one or two orders of magnitudes lower in organic particles than in dissolved form; the ratio of $^{137}$Cs activity concentration in organic particles to dissolved $^{137}$Cs activity concentration ranged from $0.01 \pm 0.00$ to $0.12 \pm$ $0.01$, except at a few stations. The ratio of dissolved $^{137}$Cs to dissolved $^{134}$Cs activity concentration changed dramatically

because of mixing with open-ocean water; it tended to increase with increasing distance from shore and to increase as the activity concentration of dissolved $^{137}$Cs decreased. In contrast, the ratio of $^{137}$Cs to $^{134}$Cs activity concentration in organic





particles did not change with distance from shore or with $^{137}$Cs activity concentration and generally remained at ~1, even at locations far from the coast. This pattern indicated that the source of the organic particles was the rivers or another source very close to the coast.

A simple mixing model for the $^{137}$Cs data obtained along the Ukedo transect on 17 October 2014 indicated that the $^{137}$Cs activity concentration at the mouth of the Ukedo River should be ~260 Bq m$^{-3}$, which is in good agreement with the activity concentration of 230 Bq m$^{-3}$ recorded in the TEPCO monitoring data at the Ukedo port on 16 October 2014.

The relationship between the decay-corrected $^{137}$Cs and $^{134}$Cs activity concentration corrected to the time of the accident showed that the pre-Fukushima $^{137}$Cs activity concentration due to global fallout were $1.3 \pm 0.2$ Bq m$^{-3}$ at sea area north of

FNPP1 region and $1.2 \pm 0.7$ Bq m$^{-3}$ at Tomioka port, respectively. These $^{137}$Cs activity concentrations are in agreement with the pre-Fukushima $^{137}$Cs activity concentration (Aoyama et al., 2008). The $^{137}$Cs/$^{134}$Cs activity ratio in dissolved form in the SoSo project samples collected north of FNPP1 was estimated to be $1.074 \pm 0.015$ which is in good agreement with the $^{137}$Cs/$^{134}$Cs activity ratio of 1.06 (+-10%) of the radiocaesium in the core of Unit 1 of the FNPP1, while it at Tomioka port was $0.998 \pm 0.017$. In the Tomioka River catchment, the source of radiocaesium might be the core of Unit 2 of FNPP1 (Nakajima

et al., 2017), then the $^{137}$Cs/$^{134}$Cs activity concentration ratio might be 0.92 (+-10%) (Nishihara et al., 2012). Therefore we concluded that the source of radiocaesium in seawater in the coastal region north of FNPP1 was deposited radiocaesium released from the core of Unit 1 of FNPP1, while the source of radiocaesium observed in the coastal region south of FNPP1 was a mixture of deposited radiocaesium released from the core of Unit 2 and the core of Unit 1 of FNPP1.

$^{137}$Cs activities at Fukushima coast and 7-day modified API showed a good positive relationship with the exception at the 56N

canal of the FNPP1. The reason for the lower correlation coefficient between $^{137}$Cs activity concentration in the 56N canal of the FNPP1 and the 5–7 day modified API might reflect some very high $^{137}$Cs activities with not heavy rain event.

**Data availability**

Activity concentrations of $^{134}$Cs and $^{137}$Cs in dissolved form, all particles, and organic particles collected during the SoSo 5

Rivers cruise activities used in this study are in a published dataset entitled "Dataset of 134Cs and 137Cs activity concentration concentrations in dissolved for, all particles and organic form of particles obtained by SoSo 5 rivers cruise in 2014" as doi: 10.34355/CRiED.U.TSUKUBA.00030 (Aoyama et al., 2020a).

Activity concentrations of $^{134}$Cs and $^{137}$Cs in dissolved form and all particles collected at 12 stations off Tomioka on 28 August 2014 used in this study are in a published dataset entitled "Dataset of 134Cs and 137Cs activity concentrations in dissolved

form and, all particles at Tomioka, Fukushima in August 2014" as doi: 10.34355/CRiED.U.TSUKUBA.00031 (Aoyama et al., 2020b).

Activity concentrations of $^{134}$Cs and $^{137}$Cs in dissolved form collected at Tomioka port during the period from 10 June 2014 to 24 April 2019 used in this study are in a published dataset entitled "Dataset of time series of radiocaesium activity concentrations at Tomioka, Fukushima during the period from 10 June 2014 to 24 April 2019" as doi:

10.34355/CRiED.U.TSUKUBA.00032 (Aoyama et al., 2020c).



**Author contribution**: SC, CD and MA designed the experiment of SoSo 5 river cruises in 2014 and carried them out. MA designed the long term experiment at Tomioka from 2014 to 2019 and carried them out. YH measured radiocaesium at Ogoya Underground laboratory for most of the samples in this study.

**Competing interests**

The authors declare that they have no conflict of interest

**Acknowledgments**

This work was part of the AMORAD project (French State financial support managed by the National Agency for Research allocated in the "Investments for the Future" framework programme as project number ANR-11-RSNR-0002). The authors thank Mireille Arnaud (IRSn) and Hervé Thébault (IRSN) for their helpful and expert assistance in field work. The authors also thank the KANSO team as well as Franck Giner (IRSN) for their helpful and expert assistance in field work. This work was also a part of a project founded by IER, Fukushima Univ. to Michio

AOYAMA.

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





Table 1 Ratios of $^{137}$Cs/$^{134}$Cs activity in cores of the FNPP1 and in surface water at the SoSo 5 rivers cruise region, north of FNPP1 and Tomioka port, south of FNPP1 and pre-Fukushima $^{137}$Cs activity concentrations.

| Location | $^{137}$Cs/$^{134}$Cs activity ratio | | | pre-Fukushima $^{137}$Cs Bq m-3 | data source |
|---|---|---|---|---|---|
| Unit 1 core | 1.06 | +- | 10% | | Nishihara et al., 2012 |
| Unit 2 core | 0.92 | +- | 10% | | Nishihara et al., 2012 |
| Unit 3 core | 0.96 | +- | 10% | | Nishihara et al., 2012 |
| | | | | | |
| Results of Standardized Major Axis regression | | | | | |
| SoSo 5 rivers cruise, north of FNPP1 | 1.074 | +- | 0.015 | 1.3 +- 0.2 | this study |
| Tomioka port, south of FNPP1 | 0.998 | +- | 0.017 | 1.2 +- 0.7 | this study |

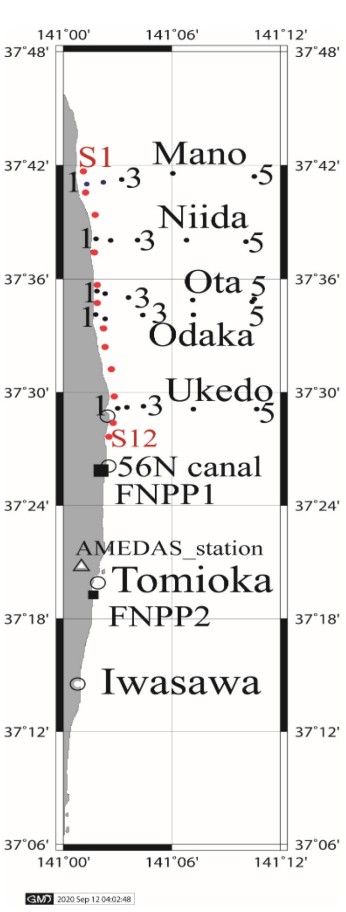

Fig. 1. Locations of stations of seawater samples in this study at 25 stations along five radial transect off the mouth of Mano, Nitta, Ota, Odaka, and Ukedo rivers (black solid circle), 12 stations from S1 to S12 along coast (red solid circle) and at Tomioka port (open circle). Four TEPCO monitoring sites of surface seawater at Ukedo port, 56 north canal of Fukushima Dai-ichi Nuclear Power Plant and Iwasawa beach (open circles) and FNPP2 port (solid square). A location of precipitation-observation, Automated Meteorological Data Acquisition System, AMEDAS, station, at Tomioka town (open triangle).





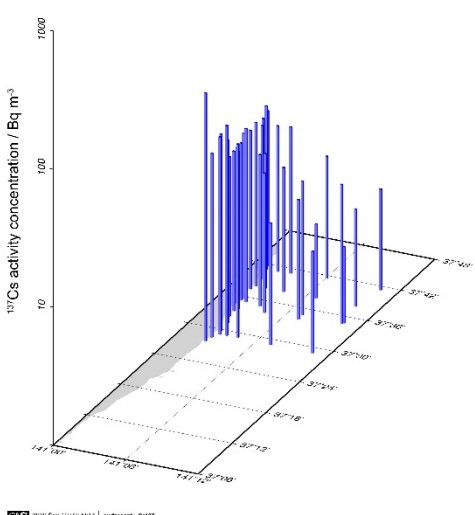

Fig. 2. 3D bar graph of $^{137}$Cs activities in dissolved form in the surface layer during October 2014 at all stations.


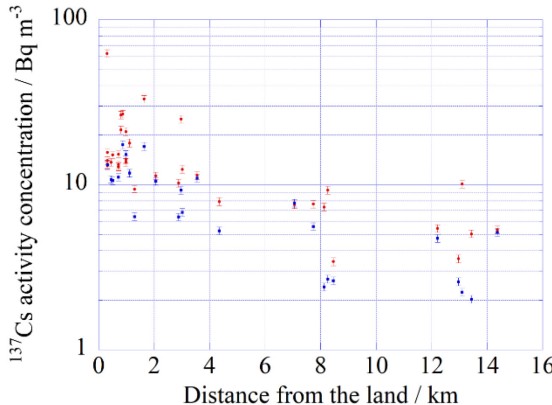

Fig. 3. $^{137}$Cs activity concentrations in dissolved form in surface and bottom waters along five radial transect off the mouth of Mano, Nitta, Ota, Odaka, and Ukedo rivers and those in surface water from S1 to S12 stations. X-axis is distance from the coast to the stations. Red solid circle: Surface water, Blue solid square: Bottom water





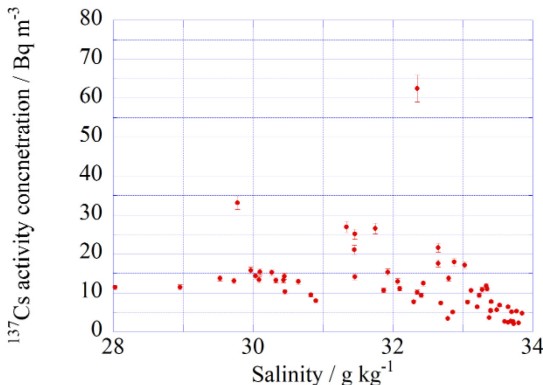

Fig. 4. Relationship between salinity and $^{137}$Cs activity concentration in all samples.

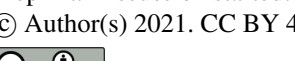



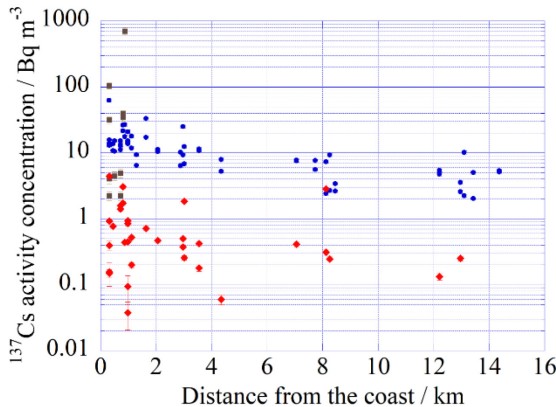

Fig. 5. Relationship between the distance from the coast and $^{137}$Cs activityconcentrations in dissolved form (blue solid circle), all particle (brown solid square) and organic form of particle (red solid star).





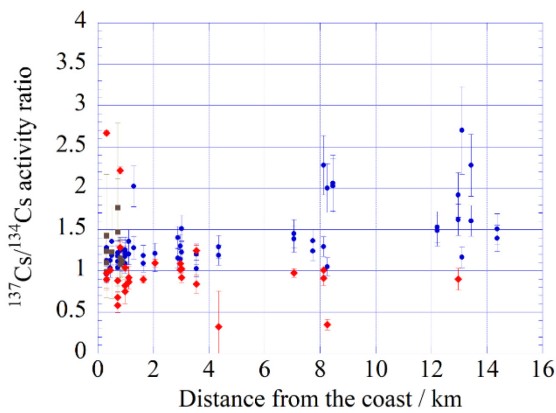

Fig. 6. Relationship between the distance from first stations on radial transects and ratios of $^{137}$Cs to $^{134}$Cs activity concentration in dissolved form (blue solid circle), all particle (brown solid square) and organic form of particle (red solid star).



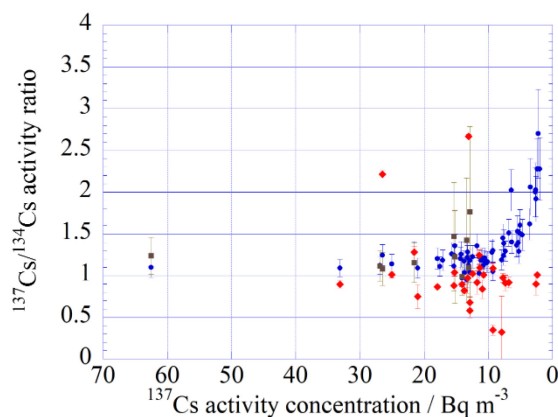

Fig. 7. Relationship between $^{137}$Cs activity concentration and ratios of $^{137}$Cs to $^{134}$Cs activity concentration in dissolved form (blue solid circle), all particle (brown solid square) and organic form of particle (red solid star).





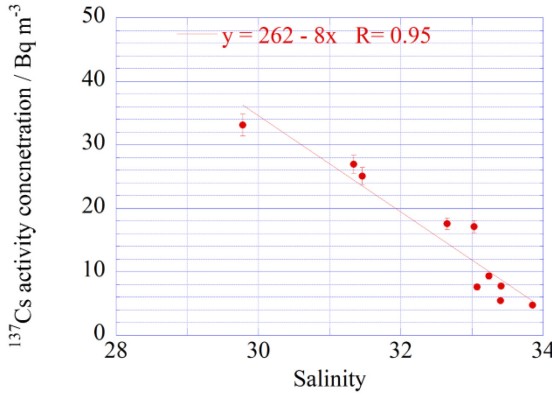

Fig. 8. Relationship between salinity and $^{137}$Cs activity concentration along only the Ukedo radial transect.



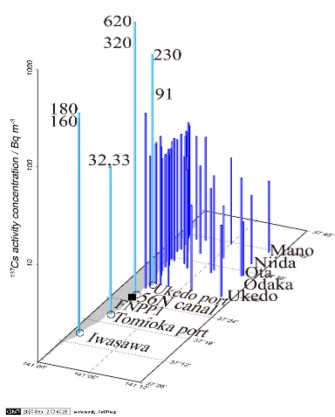

Fig. 9. 3D bar graph of dissolved [137]Cs activity in the surface water during October 2014 for all the stations in this study (dark blue bar) and TEPCO monitoring data at Ukedo port, the 56N canal at the FNPP1, and Iwasawa beach (light blue bar) observed on 8 and 16 October 2014 as well as a datum from Tomioka port (light blue bar) on 3 and 13 October 2014. The two numbers for each light blue bar are [137]Cs activity in the surface water at each station.



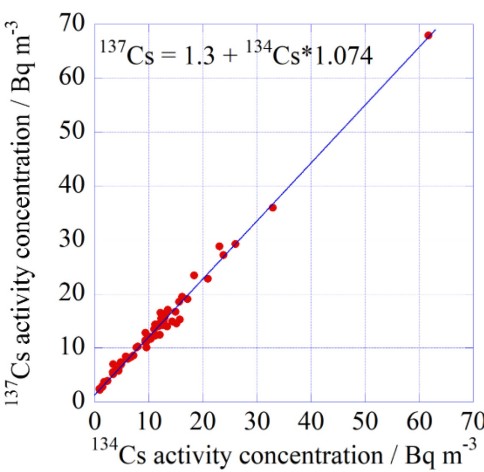

Fig. 10. Relationship between [137]Cs and [134]Cs activity concentration in dissolved form which are decay corrected on 11 March 2011 in the SoSo 5 rivers cruise data.





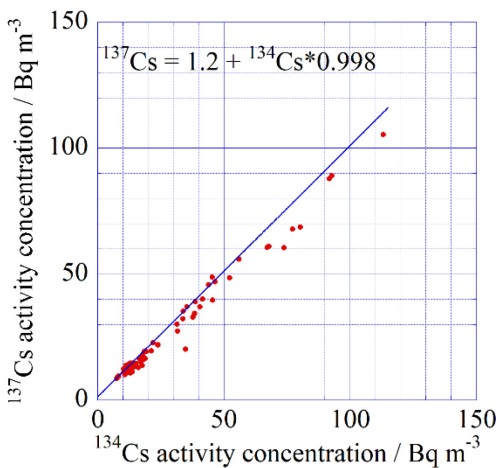

Fig. 11. Relationship between $^{137}$Cs and $^{134}$Cs activity concentration in dissolved form which are decay corrected on 11 March 2011 at Tomioka port during the period from June 2014 to April 2019.





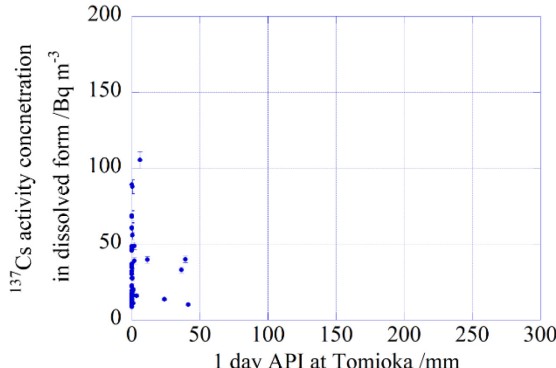

Fig. 12. Relationship between 1 day modified API and the [137]Cs activity concentration in surface water at Tomioka port.



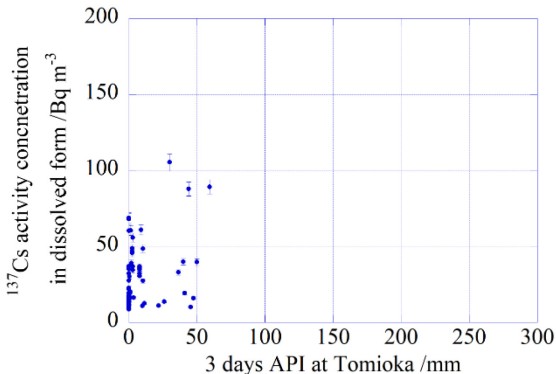

Fig. 13. Same as Fig. 12 but for 3 days modified API.





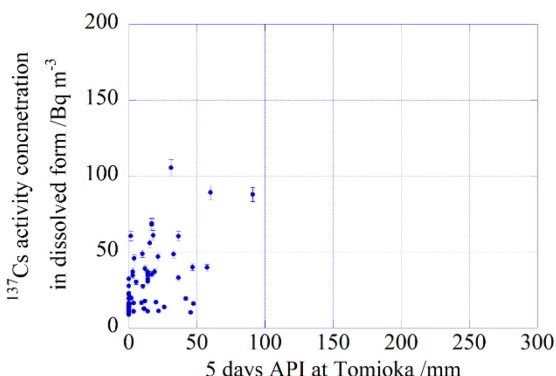

Fig. 14. Same as Fig. 12 but for 5 days modified API.





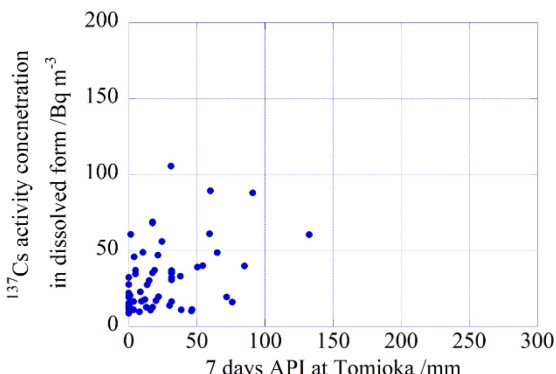

Fig. 15. Same as Fig. 12 but for 7 days modified API.





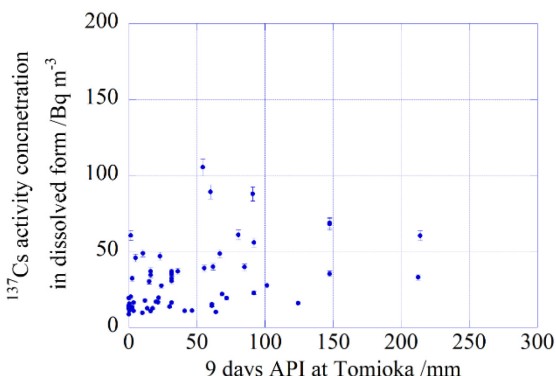

Fig. 16. Same as Fig. 12 but for 9 days modified API.





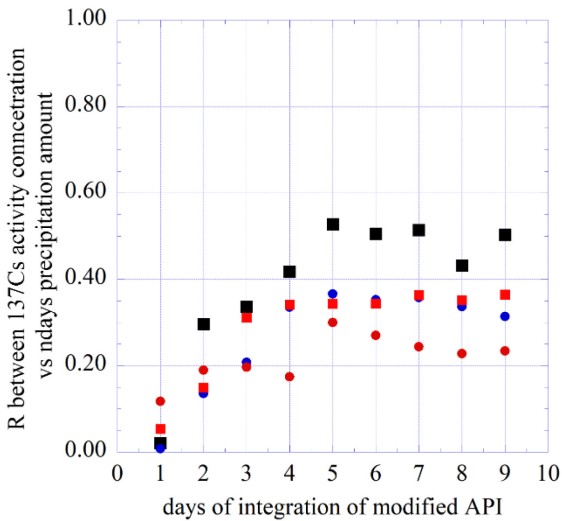

Fig. 17.  Correlation coefficients between the 1–9-day modified API and $^{137}$Cs activity concentrations at Tomioka port (black solid square) , Ukedo port (blue solid circle), the 56N at FNPP1 (red solid circle) , and Iwasawa beach (red solid square).





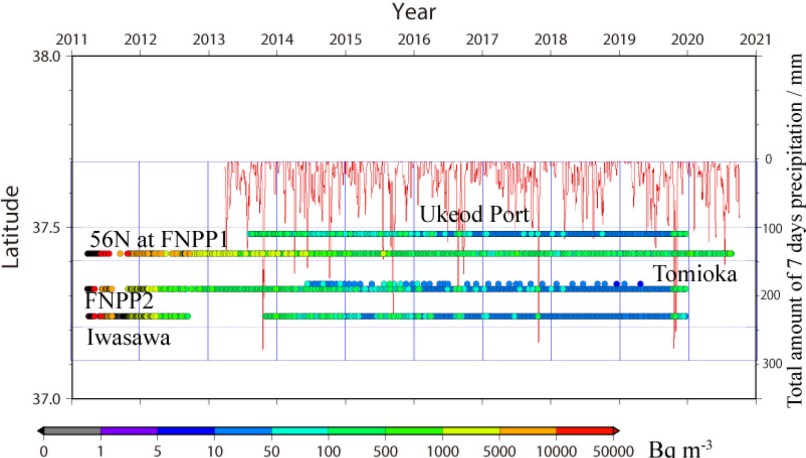

Fig. 18. Hovmöller diagram of the $^{137}$Cs activity concentrations at Ukedo port, the 56N of the FNPP1, Tomioka port, at FNPP2 port and Iwasawa beach. The 7-day modified API with a reversed ordinate.





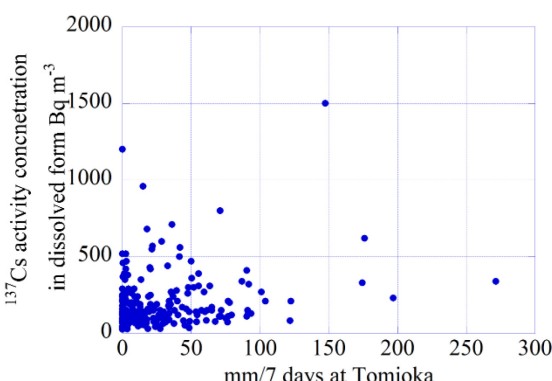

Fig. 19. Relationship between [137]Cs activities at the 56N canal of the FNPP1 and 7-day modified API.