# Peer review of "Impact of typhoons on particulate and dissolved 137Cs activities in seawater off the Fukushima Prefecture: results from the SOSO 5 Rivers cruise (October 2014)"

_Biogeosciences, 2020_

## Author Comment (AC1)

**Replies to Comment on bg-2020-491 by anonymous referee #1**

**Dear anonymous referee #1,**

**We submit our replies to your comments as below.**

**Thank you very much for your comments which greatly contribute to the improvement of our manuscript.**

**Best regards,**

**Michio**

\*\*\*\*\*\*\*\*\*\*\*\*\*\*\*\*\*\*\*\*\*\*\*\*\*\*\*\*\*\*\*\*\*\*\*\*\*\*\*\*\*\*\*\*\*\*\*\*\*\*\*\*\*\*\*\*\*\*\*\*\*\*

This work presents radiocesium in water and particles collected from the areas off the mouths of five rivers north of Fukushima Dai-ichi Nuclear Power Plant (FDNPP1) shortly after the passage of two typhoons. As the decrease of local atmospheric deposition and the direct discharge of radiocesium from the reactor, rive runoff input of radiocesium into the Pacific Ocean might become an important contribution. However, data presented in work did not give a clear picture about the impact of typhoon on particulate and dissolved 137Cs activities in seawater off the Fukushima Prefecture. The quality of writing, data presentation and results discussion do not meet the criteria to be accepted as a research article on Biogeosciences.

**A: Ok, we change the title from "Impact of typhoons on particulate and dissolved 137Cs activities in seawater off the Fukushima Prefecture: results from the SOSO 5 Rivers cruise (October 2014)" to "Behavior of particulate and dissolved 137Cs activities in seawater off the Fukushima Prefecture: results from the SOSO 5 Rivers cruise (October 2014)"**
**We will revise our manuscript based on RC1's comments except for discussion on relationships between 137Cs activity concentration in dissolved form and 1day to 9 days API. We also believe that our results deserve to be published in Biogeosciences and that the revised version will allow us to better highlight them.**

General comments:

- The quality of figures in this work are very poor. Some of the figure are hardly to see the values, and the trends as the authors described in the context, especially on paper print. Some of the figures can be plotted in a different way to make them more illustrative.

  **A: OK. We will improve the quality of figures.**

- Novelty and Objective. It is not very clear in the introduction what is the scientific question in this work? The novelty is not highlighted either, what is new in this study? Why the authors carried out the investigation to exam the radiocesium in water and particles? The sentence in line 86-87 is very confusing. The authors mentioned 'to understand riverine fluxes in the coastal region' but in the discussion on the 'river fluxes' and its connection to precipitation is very weak. It is still not clear how significant these 'river fluxes' contribute the total inventory of radiocesium in the study area.

  **A: New findings and novelty in our article are shown below.**

1, We highlighted simple two end-members mixing between river water and open sea water after flooding due to heavy rain showed by data in Fig. 8, Salinity vs. $^{137}$Cs activity concentration. Extrapolation to salinity equal zero gave us an appropriate number of $^{137}$Cs activity concentration in dissolved form at the mouth of the Ukedo river. This is the first data set of this kind.

2, Contrary to the dissolved phase, $^{137}$Cs/134Cs ratio decay corrected to 11 march 2011 in the extracted organic fraction (extracted by conc. H2O2) did not change with distance from the coast underlying the decoupling of these two phases Dissolved phase follows a classical two end-member mixing while the extracted organic phase is characterized by ratio of the FDNNP1 releases even at distance from the coast

3, The amount of precipitation for the day and $^{137}$Cs activity concentration were not correlated as shown in Fig. 12 (fig 17 see comment on figure 12-16 and 19 later in the text below), 1day API vs. $^{137}$Cs activity concentration which clearly indicated that a simple saying like "heavy rain is a cause of higher $^{137}$Cs activity concentration" is far from reality.  Indeed, a timelag of 10 to 60 hours is generally observed between the time of the rains and the increase in the (total) flow of the rivers depending on the watershed scale including the magnitude of the slope gradient of the Tohoku region in Japan. In addition, the main inputs of radiocaesium from land to the ocean are via riverine particle fluxes (Sakuma et al., 2019). It is therefore necessary to consider also the delay due to the phase change of part of the caesium fixed on the particles towards the dissolved phase at river mouth due to change in physico-chemical conditions, especially ionic strength. According to Delaval et al (2020) two first-order reactions govern the kinetics of the process, with half-life reaction times of 1 h and a few days depending on the sites of cesium binding on the particles.
Therefore, a delay of a few to several days can be observed between the time of heavy rains and the increase of dissolved cesium in the coastal area in connection with biogeochemical and hydrological processes.
These background knowledges underpin our finding that the dissolved radiocaesium activity concentration shows a correlation with several days API, and not with the amount of the precipitation of the day.

Objective of this work is the study of the behavior of particle and dissolved radiocaesium at coastal region of Fukushima based on detailed observations.

RC1 asks that "It is still not clear how significant these 'river fluxes' contribute the total inventory of radiocesium in the study area.". Flux estimation is not an aim of this paper and we have submitted an article to Biogeoscience "Aoyama, M., Charmasson, S., Hamajima, Y., Duffa, C., Tsumune, D., and Tateda, Y.: Tritium activity concentration and behaviour in coastal regions of Fukushima in 2014, Biogeosciences Discuss. [preprint], https://doi.org/10.5194/bg-2021-10, in review, 2021.". In this article we have presented estimates of 3H and 137Cs fluxes from rivers located north and south of FNPP1, FNPP1 and open water (see below).

| Table 3 Estimated flux of 3H and 137Cs | | | |
|---|---|---|---|
| | June 2014 | October 2014 | October 2019 |
| | Bq day-1 | Bq day-1 | Bq day-1 |
| **Total 3H flux** | **5.8E+10** | **6.3E+10** | **6.1E+10** |
| Open water | 5.2E+10 | 5.2E+10 | 5.2E+10 |
| Rivers north of FNPP1 | 3.2E+09 | 5.3E+09 | 6.0E+09 |
| The port of FNPP1 | 1.9E+09 | 4.5E+09 | 1.4E+09 |
| Two rivers south of FNPP1 | 4.2E+08 | 6.6E+08 | 1.0E+09 |
| | | | |
| 56N at FNPP1(not included to total) | 8.6E+10 | 5.6E+10 | 2.8E+10 |
| | | | |
| **Total 137Cs flux** | **5.0E+09** | **4.5E+09** | **2.8E+09** |
| Open water | 1.6E+09 | 1.6E+09 | 1.6E+09 |
| Rivers north of FNPP1 | 5.9E+08 | 7.4E+08 | 1.9E+08 |
| The port of FNPP1 | 2.8E+09 | 2.1E+09 | 9.7E+08 |
| Two rivers south of FNPP1 | 1.4E+08 | 1.6E+08 | 5.4E+07 |
| | | | |
| 56N at FNPP1(not included to total) | 8.4E+09 | 8.0E+09 | 3.4E+09 |

- The definition of organic particle. The authors used concentrated nitric acid and hydrogen peroxide to process the samples and claim the obtained radiocesium was associated to organic form. I disagree to term this fraction as 'organic particle', because
1) the experiment details were not clear. What was the ratio between sample and reagents, what temperature was applied? 2) There was no data/literature to support the obtained radiocesium was only associated to organic matter. As concentrated nitric acid is a strong acid, it can easily dissolve inorganic radiocesium absorbed on the particle surface and can also extract radiocesium incorporated to the mineral lattices. Even though the authors reported the results of radioacesium for this 'organic particle' fraction, the interpretation and discussion on the results is not in depth. Why the authors analyzed radiocesium in 'organic particles'? Why 137Cs activities concentration in 'organic particles' were one or two order lower than dissolved 137Cs (line 173), while 137Cs in all particles were higher than dissolved 137Cs (line 169)? How these results are connected to typhoon?

**A: In the current text I made a copy paste mistake. We only used conc. H2O2 with pure water, and did not use conc. HNO3. The Actual method for getting the organic fraction is given below.**

**On the filters there were yellow-brown matters for all surface layer samples. Then we poured H2O2 step by step about 2 ml each time. We stopped adding H2O2 when the color disappeared.** The fraction dealt with in this paper corresponds to this soft extraction. **We revised this part as appropriately as below.**

**In the current text, we stated that "Organic form of radiocaesium of the samples were obtained by disillusion of organic portion on the filter using concentrated nitric acid and concentrated hydrogen peroxide at all stations, then filled and dried up in a Teflon tube.".**

**We will revise as below based on our lab note by my staff.**

**We performed filtration using membrane filter with a pore size of 0.45-μm (Millipore HA). The filters were dried up and weighed to determine the mass of particles on the filter. Then, 4ml conc. H2O2 were poured on the filter placed in a plastic try three times with an interval of a few days, 5 Dec.2014, 8 Dec. 2014, 10 Dec. 2014 for all treated filters. And 4ml pure water were added on 8 Dec. 2014. Then if orange or brown parts still exist, we added pure water 4ml and 4ml con.H2O2 again on 15 Dec. 2014 to about half the number of filters. The end point was the visual inspection of the color disappearance. The solutions were put in a Teflon tube and measured at Ogoya laboratory after drying. Therefore, in**

**the revised article we will mention extracted organic fraction instead of organic particles**

- 137Cs/34Cs ratio in particles. The authors in section 4.2 discuss the source term of 137Cs based on the 137Cs/134Cs in seawater, but why the authors did not discuss the obtained results for 137Cs/134Cs in particles? The authors claim the 137Cs/134Cs ratios in particle do not changes with distances but what are these values obtained in this work? Do they agree with the estimated ratios for FDNPP? Again, how typhoon impact the distribution of 137Cs/134Cs ratios?

    **A: As shown in Fig 6, the 137Cs/134Cs ratio in particles was only obtained for the very coastal region and we do not have data to study the link with distance given the very small amount of material collected outside the area very close to the coast preventing us from carrying out gamma spectrometry. However, the new finding is that the 137Cs/134Cs ratio in the organic fraction extracted remains close to 1 and does not change with distance.**

    **137Cs/134Cs activity ratio in particles are shown in Figs.6 and 7. General tendency is same as 137Cs/134Cs activity ratio in the extracted organic fraction. We will add a few sentences on 137Cs/134Cs activity ratio in particles. We do not have data on distance dependency for all particles. (see new fig. 6)**

    **We stated that "the 137Cs/134Cs ratios in the extracted organic particle do not changes with distances" and not that "the 137Cs/134Cs ratios in particle do not changes with distances". The data are in DOI00030 and Figs. 6 and 7.**

- API calculation. I am not convinced by the API calculation approach. As the definition of K is not clear, what is the difference between K1, K2, …Kn, Why the authors set K to 1? Besides, from Fig. 17, the maximum value for R is about 0.5, meaning the R2=0.25. Therefore, none of these correlations is significant. I do not think it make much sense to perform such correlation analysis. Direct use of the metrological data (e.g., rainfall) would be sufficient to support the conclusion that high 137Cs activity concentrations in September-October were connected to the typhoon events.

    **A: It is clear that there is no relationship between the amount of rain of the day and the 137Cs activity. It is therefore not scientific just to say that high 137Cs activity concentrations in September-October were connected to the typhoon events. By drawing time series of precipitation and 137Cs activity, anyone can see good relationships between these two variables, but this is only visual based. If we want to go more deeply in these relationships we have to take into account the fluvial phenomena, that is to say we must consider Hydrology and biogeochemistry (see sentences #3** in reply to **Novelty and Objective) and how 137Cs activity is connected/related to with the precipitation in the watershed and phase change from particle to dissolved form when entering the marine environment. So in this article we looked at relationship between the API and 137Cs activity concentration. This constitutes one of the novelties of this article.**

    **For API calculation, when we could not get detailed hydrological information on interested region, it is common to use all K=1.**

    **As RC1 pointed out, the correlation coefficients may appear weak. But the amount of data is substantial and we tested the significance of these coefficients by a student's t test. The API 4 days to the API 9 days were significantly correlated with the activity concentration of [137]Cs at a confidence level of 99% as indicated below.**

| station | | number of data | 1day API | 2days API | 3days API | 4days API | 5days API | 6days API | 7days API | 8days API | 9days API |
|---|---|---|---|---|---|---|---|---|---|---|---|
| Tomioka | n= | 72 | r= | -0.0208 | 0.2964 | 0.3362 | 0.4177 | 0.5283 | 0.5059 | 0.5139 | 0.4322 | 0.5033 |
| | | | t | 0.1737 | 2.5967 | 2.9869 | 3.8464 | 5.2053 | 4.9069 | 5.0124 | 4.0100 | 4.8733 |
| | | | p | 0.8626 | 0.0115 | 0.0039 | 0.0003 | 0.0000 | 0.0000 | 0.0000 | 0.0001 | 0.0000 |
| | | | | | | | | | | | | |
| Ukedo | n= | 291 | r= | -0.0077 | 0.1358 | 0.2075 | 0.3352 | 0.3657 | 0.3532 | 0.3577 | 0.3373 | 0.3140 |
| | | | t | 0.1305 | 2.3295 | 3.6058 | 6.0493 | 6.6789 | 6.4170 | 6.5122 | 6.0913 | 5.6220 |
| | | | p | 0.8962 | 0.0205 | 0.0004 | 0.0000 | 0.0000 | 0.0000 | 0.0000 | 0.0000 | 0.0000 |
| | | | | | | | | | | | | |
| Iwasawa | n= | 243 | r= | 0.0536 | 0.1497 | 0.3114 | 0.3412 | 0.3433 | 0.3441 | 0.3634 | 0.3516 | 0.3651 |
| | | | t | 0.8329 | 2.3511 | 5.0876 | 5.6342 | 5.6741 | 5.6902 | 6.0549 | 5.8298 | 6.0876 |
| | | | p | 0.4057 | 0.0195 | 0.0000 | 0.0000 | 0.0000 | 0.0000 | 0.0000 | 0.0000 | 0.0000 |
| | | | | | | | | | | | | |
| 56N | n= | 243 | r= | 0.1175 | 0.1899 | 0.1965 | 0.1740 | 0.3005 | 0.2701 | 0.2437 | 0.2280 | 0.2342 |
| | | | t | 1.8361 | 3.0025 | 3.1106 | 2.7437 | 4.8907 | 4.3542 | 3.9016 | 3.6350 | 3.7401 |
| | | | p | 0.0676 | 0.0030 | 0.0021 | 0.0065 | 0.0000 | 0.0000 | 0.0001 | 0.0003 | 0.0002 |

- The language shall be improved thoroughly, as many descriptions are not concise. There are also repeated contexts.

   **A: Ok these various aspects will be improved.**

Specific comments:

- Line 37: please give uncertainties for these 137Cs/134Cs ratios.
-

   **A: Nishihara et al., 2012 did not give uncertainties on inventory estimates in the three cores, but around 10 % should be a fair number of uncertainties for these 137Cs/134Cs ratios uncertainties based on personal communication with Dr. Nishihara. In Table 1, we already put 10% as magnitude of uncertainty to these ratios.**

   Line 40: why the measurements were inaccurate? Please explain.

   **A: The sentence was incomplete.**

   **In the current text we stated as below.**

   **In the coastal region, however, the $^{137}$Cs to $^{134}$Cs activity concentration ratio in dissolved radiocaesium in seawater was reported to be uniform and very close to 1 (Buesseler et al., 2012;Buesseler et al., 2011), probably because of inaccuracies in the measurements made during monitoring by the Japanese Government and Tokyo Electric Power Company Holdings, hereafter TEPCO.**

   **We will revise this part as below**

   **In the coastal region, however, the $^{137}$Cs to $^{134}$Cs activity concentration ratio in dissolved form in seawater was reported to be uniform and very close to 1 (Buesseler et al., 2012;Buesseler et al., 2011). When we look at the TEPCO and governmental monitoring data, the 137Cs to 134Cs activity concentration ratios were scattered around 1 and could not distinguish the ratios originated from core 1 and core2/3. This probably because of larger uncertainty and inaccurate measurements made during monitoring and also mixing processes along the coast among releases originating from these various cores. Up to now, there is no report on 137Cs to 134Cs activity concentration ratios in seawater in which two different ratios were distinguished by observations.**

- Line 43-49: these are earlier findings about the different isotopic signatures on the

particles affected by different units of FDNPP. The summary should be shortened to focus on the key point for this work.

**A: OK we do so.**

**In the current text we stated as below.**
**Miura et al. (Miura et al., 2020) have reported two types of caesium-bearing microparticles emitted from the FNPP1 accident that were separated from road dust and non-woven fabric cloth. Type-A particles, which were spherical and ~0.1–10 μm in diameter, contained ~$10^{-2}$ to $10^2$ Bq of $^{137}$Cs radioactivity concentration. Type-B particles, which had various shapes and were 50–400 μm in diameter, contained $10^1$–$10^4$ Bq of $^{137}$Cs radioactivity concentration. The $^{137}$Cs to $^{134}$Cs activity concentration ratios in Type A and Type B particles were in excellent agreement with the corresponding ratios in cores 2 and 3 in the former case and in core 1 in the latter case. The evidence therefore indicated that the type A particles came from Units 2 and 3, and the Type B particles came from Unit 1.**

**We will revise as below,**

**Miura et al. (Miura et al., 2020) reported two types of caesium-bearing microparticles emitted during the FNPP1 accident and the $^{137}$Cs to $^{134}$Cs activity concentration ratios in these two types of particles were in excellent agreement with the corresponding ratios in cores 2 and 3, and in core 1, respectively.**

- Line 62-74. These are also very tedious literature review. It is better to extract the most important findings which are relevant to this work.

  **A: We do not think that this is a "very tedious review" as the assessment of 137Cs river fluxes is very important and we want to give good information about this issue. But, we will make this part more concise.**

- Line 157-159: these values are not reflected anywhere in the figures or table. These sentences should also be more concise.

  **A: readers can easily calculate these values in line157-159 based on observed data as shown in DOI00030. We will make more concise here.**

- Line 162-165. These are somehow repeated sentences.
  **A: yes, we will revise as you suggested.**

- Line 187-189: it is a very long sentence, please modify. Besides, it is not clear whether the findings about organic particles in this work agree with Naulier et al.

  **A: It is true that we do not have data on both the total suspended matter and the organic fraction extracted which would have allowed a more complete comparison with the work by Naulier et al. However, the fact that the 137Cs / 134Cs ratios are different in the dissolved phase and in the fraction extracted underlines a decoupling of these two phases. These ratios show that the extracted organic fraction is labelled by the releases from the FDNNP1 even at distance from the coast contrary to the water masses containing them, which corroborates the importance of the contributions of organic matter in the transfer of cesium from land to the sea.**

- The conclusion should be re-written. The conclusion shall not compile all the detailed results obtained from the work. It shall be more informative to let the reader understand the main points of the work and provides the reader with a sense of closure on the topic.

  **A: We thought it was important in articles of this type based on observations to have a detailed and precise conclusion. However, we will revise the conclusion section, taking into account RC1's comments, to clarify the content and highlight the novelty and new findings of our article for readers. Although these changes are provisional and will be revised again, some sentences for conclusion which will be included in the upcoming revised article are shown below to meet your request.**

**The detection of both 134Cs and 137Cs in all dissolved samples demonstrated contamination from the FNPP1 accident with higher values in the surface layer compared to the bottom layer and a clear decrease with**

distance from the coast. The decrease of caesium activity concentration with increasing salinity reflected mixing of coastal water with open-ocean water.

At the stations very close to the coast relatively high 137Cs activities in all particles with values exceeding the dissolved 137Cs activities were observed. Beside, 137Cs activities were generally one or two orders of magnitudes lower in the extracted organic fraction than in the dissolved fraction reflecting the fact that the organic component of the particulate matter sampled is not the main carrier of cesium.

The 137Cs/134Cs activity concentration ratio in the dissolved phase changed drastically due to mixing with open-ocean water. In contrast, this ratio in the extracted organic fraction did not change with distance from shore or with 137Cs activity concentration and generally remained close to 1, even at locations far from the coast. This pattern indicates a decoupling between these two phases with the extracted organic fraction characterizing particle originating from land even at distance from the coast.

When considering 137Cs/134Cs ratios, the source of radiocaesium in the coastal region north of FNPP1 should be related to releases from the core of Unit 1 of FNPP1, while in the coastal region south of FNPP1 the source should be a mixture of releases from the core of Unit 2 and the core of Unit 1 of FNPP1.

137Cs activities on the Fukushima coast and the modified 7-day API showed a good positive relationship with the exception of 56N canal of FNPP1. In fact, at this location, high 137Cs activities are observed apart from an event of heavy rain.

This study shows the need to better characterize the inputs of rivers in taking into account on one side their hydrologic characteristics and on the other hand the nature of their solid discharges and phase change from particle to dissolved form. Special attention should be payed to the organic fraction of the riverine inputs as well as their fate in the coastal environment in the area which has been impacted by the accident.

- 1. It is better to combine two figures, with one larger scale map indicating geographical positon and the current circulation pattern of the study area, and another details the sampling stations. It better to remove some of the annotations to the figure caption or use smaller fonts so it does not look so squeezed, and to use different symbols to mark different rivers so the reader can easily follow.6

    A: We will revise as you suggested.

- 2 and Fig. 9, they are very unclear. Why the authors do not use Ocean Data View to present the distribution of radio cesium on the surface water?

    A: In case of that when gradient of target data is high and one side boundary is land, ODV interpolation usually give us inaccurate figures, so we use 3D bar graph to show accurate distribution of 137Cs in coastal region. But, we will try to clarify these two figures in the upcoming revised article.

- Fig 3-7, the symbols are too small to be distinguished. The authors pull data for all stations in these figures, very hard to extract useful information. Why the authors do not use different symbols to present different rivers, so the reader can visualize the variations.

    A: OK. We redraw figures with 6 different marks for on one hand the 5 river transects and the other hand S1-S12 stations and show the figures here.

    In the new figures, we allocate marks as below

    Mano river transect: solid square
    Niida river transect: solid circle
    Odaka river transect: solid up-pointing triangle
    Ota river transect: slid down-pointing triangle
    Ukedo river transect: solid diamond
    Coastal stations S1 – S12 : ◣ Lower Left Triangle

    New Figure 3

[Figure]

**For 5 rivers transect, red is for surface data and blue is for bottom data.**

**New Figure 4**

[Figure]

**For 5 rivers transect, red is for surface data and blue is for bottom data.**

**New Figure 5**

[Figure]

Blue is for dissolved form of 137Cs in seawater
Red is for 137Cs in the extracted organic fraction
Dark brown is for 137Cs in all particle

**New Figure 6**

[Figure]

**Blue is for dissolved form of 137Cs in seawater**
Red is for 137Cs in the extracted organic fraction
Dark brown is for 137Cs in all particle

**New Figure 7**

[Figure]

Blue is for dissolved form of 137Cs in seawater
Red is for 137Cs in the extracted organic fraction
Dark brown is for 137Cs in all particle

- 12-16 and Fig.19, I do not see the meaning for presenting these figures. All the figure do not show any statistical analysis data, based on the visual inspection, I do not see any of them have significant correlation. I would suggest remove them.

**A: We agree to delete fig 12 to 16 and fig 19, they are indeed not very informative and redundant with fig 17 that presents in a concise manner the clear improvement of correlation when integrating amount of daily precipitation over several days. The R coefficients may seem low however due to the large number of observations they are significant**
**(see test results in the reply about API previously)**

18. It is very hard to see the color change, especially after 2014. I suggest either delete the color bar data before 2013, or add another zoom-in figure for the data after 2014.

**A: After 2014, the reality is that changes in 137Cs activity were small except during heavy flooding. We also want to emphasize that the color bar is not linear but log, so we do not need to zoom for data after 2014.**

Technical corrections.

- Line 41: 'small but significant', this is a contradictory expression. Please modify.
- Lien 55: 'primarily' should be 'primary'.
- Line 113: 'disillusion' should be 'dissolution'.
- Line 127-138. There is an overlap with the description in 'Data availability'.
- Line 150 and 155: please delete repeated data information 'xxxdoi:xxxxx', only keep the reference is sufficient.

**A: We will revise as appropriately.**

**End of reply to RC1.**

---

## Author Comment (AC2)

**Replies to Comment on bg-2020-491 by anonymous referee #2**

**Dear anonymous referee #2,**

**We submit our replies to your comments as below.**

**Thank you very much for your comments which greatly contribute to the improvement of our manuscript.**

**Best regards,**

**Michio**

**\*\*\*\*\*\*\*\*\*\*\*\*\*\*\*\*\*\*\*\*\*\*\*\*\*\*\*\*\*\*\*\*\*\*\*\*\*\*\*\*\*\*\*\*\*\*\*\*\*\*\*\*\*\*\*\*\*\*\***

Comments on "Impact of typhoons on particulate and dissolved 137Cs activities in seawater off the Fukushima Prefecture: results from the SOSO 5 Rivers cruise (October 2014) (bg-2020-491)"

Recommendation: Accept, with major revisions noted.

General comments: I reviewed the manuscript " Impact of typhoons on particulate and dissolved 137Cs activities in seawater off the Fukushima Prefecture: results from the SOSO5 Rivers cruise (October 2014) (bg-2020-491), submitted by Aoyama et al to Biogeosciences. The authors measured 134Cs and 137Cs in the dissolved and particulate samples contaminated by the Fukushima Dai-ichi Nuclear power plant (FDNPP1) accident, which presented some new data. Their spatial distribution reflected the mixing of coastal water and open-ocean water. The 137Cs/ 134Cs activity ratio derived from FDNPP accident is used to trace the source of riverine particle, which is very interesting. They also discussed the impact of typhoons on particulate and dissolved 137Cs activities in seawater off the Fukushima prefecture, but did not give a clear picture about the impact of typhoon on 137Cs activities in seawater. The novelty of this study needs to be improved. Additionally, decisions made with respect to data presentation combined with grammatical and other organizational errors result in a MS that lacks clarity and is difficult to follow. It is necessary to polish this manuscript by a native English speaker. Therefore, it is recommended to be published after major revisions.

> **A: The manuscript will be revised in highlighting the novelty of the study that is to say two end-members mixing between river water and open sea water, the decoupling between dissolved and the organic particulate fraction extracted as shown by the trend of their respective $^{137}$Cs/134Cs ratios with distance from the coast and finally the use of multi-days API that shows relationship between precipitations and $^{137}$Cs concentrations**
>
> **In addition, as suggested the manuscript will be reorganized to give it more clarity and make it easier to read. Figures will be improved as well as English**
>
> **The specific and technical comments indicated below will be taken into account.**

Specific comments
-Line 14: What is indicated by the dissolved activities.......? $^{137}$Cs or $^{134}$Cs?
**A: Both $^{137}$Cs and 134Cs in dissolved form and extracted organic form decreased with distance from the coast due to mixing between river water and open sea water.**

-Line 18, "ranged from....to ......" means a range of variation, so the uncertainty in this sentence that "the ranged from 0.01±0.00 to 0.12±0.01" is redundant? Please note this in the MS.

**A: The ratios shown here are based on two radioactivity measurements results, therefore ratios have also uncertainties.**

-Material and methods: What are the detection limits of $_{134}$Cs and $_{137}$Cs?

**A: The detection limit for net $^{137}$Cs activity is about a few mBq per sample and that for $^{134}$Cs is about 10 mBq per sample. The measurements were carried out at the Ogoya underground laboratory which is located 270 meter water equivalent deep.**

-Lines 169-171, this sentence (the ratio of particulate $_{137}$Cs activity concentration......) is confusing, please rephase it.

**A: Ok, we will do so.**

- Lines 179-181. This sentence is too long and needs revise to improve clarity and the flow....

**A: Ok, we will do so.**

-Lines 206-211, what's meaning that "data not shown or figure not shown"? Add in the Supporting information?

**A: all salinity data and radiocaesium activity concentration in dissolved form data are presented in Aoyama et al., 2020a. So, (figure not shown; data are in Aoyama et al., 2020a) at line209-201 is correct and (data not shown) at line 207 should read as (figure not shown; data are in Aoyama et al., 2020a) in the original text. By the way, due to Rc1's comments, new figure4 will be in the coming revised article. So, both statements will be replaced as (see figure4, data are in Aoyama et al., 2020a).**

[Figure]

**Fig. 4. Relationship between salinity and $^{137}$Cs activity concentration in dissolved form.**
    **Mano river transect: solid square**
    **Niida river transect: solid circle**
    **Odaka river transect: solid up-pointing triangle**
    **Ota river transect: slid down-pointing triangle**
    **Ukedo river transect: solid diamond**
    **For 5 rivers transect, red is for surface data and blue is for bottom data.**

-Discussion section: The discussion was not enough and some conclusions are soft or from conjecturing, for example, "this pattern might reflect complex physical processes....."(lines 210-211); "Possible explanation of this finding are that the radiocaesium in the coastal seawater........"(Lines 236-239).

**A: We will revise the discussion and conclusion based on the statement regarding with novelty of this article. New findings and novelty in our article are shown below.**

**1, We highlighted simple two end-members mixing between river water and open sea water after flooding due to heavy rain showed by data in Fig. 8, Salinity vs. $^{137}$Cs activity concentration. Extrapolation to salinity equal zero gave us an appropriate number of $^{137}$Cs activity concentration in dissolved form at the mouth of the Ukedo river. This is the first data set of this kind.**

**2, Contrary to the dissolved phase, $^{137}$Cs/134Cs ratio decay corrected to 11 march 2011 in the extracted organic fraction (extracted by conc. H2O2) did not change with distance from the coast underlying the decoupling of these two phases Dissolved phase follows a classical two end-member mixing while the extracted organic phase is characterized by ratio of the FDNNP1 releases even at distance from the coast**

**3, The amount of precipitation for the day and $^{137}$Cs activity concentration were not correlated as shown in Fig. 12 (fig 17 see comment on figure 12-16 and 19 later in the text below), 1day API vs. $^{137}$Cs activity concentration which clearly indicated that a simple saying like "heavy rain is a cause of higher $^{137}$Cs activity concentration" is far from reality. Indeed, a timelag of 10 to 60 hours is generally observed between the time of the rains and the increase in the (total) flow of the rivers depending on the watershed scale including the magnitude of the slope gradient of the Tohoku region in Japan. In addition, the main inputs of radiocaesium from land to the ocean are via riverine particle fluxes (Sakuma et al., 2019). It is therefore necessary to to consider also the delay due to the phase change of part of the caesium fixed on the particles towards the dissolved phase at river mouth due to change in physico-chemical conditions, especially ionic strength. According to Delaval et al (2020) two first-order reactions govern the kinetics of the process, with half-life reaction times of 1 h and a few days depending on the sites of cesium binding on the particles. Therefore, a delay of a few to several days can be observed between the time of heavy rains and the increase of dissolved cesium in the coastal area in connection with biogeochemical and hydrological processes.**
**These background knowledges underpin our finding that the dissolved radiocaesium activity concentration shows a correlation with several days API, and not with the amount of the precipitation of the day.**

-Conclusions section: The conclusion section seems long with too much information on some discussion that appears unnecessary. The conclusion should be rephase.

**A: Yes, we will do it. RC1 also gave us similar comments. We will revise the conclusion section taking into account your comments as well as those of RC1, in order to make it clearer and better underline the results of this study. While these changes are provisional and will be revised again, some concluding paragraphs which will be included in the upcoming revised article are presented below to meet your request.**

**The detection of both 134Cs and $^{137}$Cs in all dissolved samples demonstrated contamination from the FNPP1 accident with higher values in the surface layer compared to the bottom layer and a clear decrease with distance from the coast. The decrease of caesium activity concentration with increasing salinity reflected mixing of coastal water with open-ocean water.**

**At the stations very close to the coast relatively high $^{137}$Cs activities in all particles with values exceeding the dissolved $^{137}$Cs activities were observed. Beside, $^{137}$Cs activities were generally one or two orders of magnitudes lower in the extracted organic fraction than in the dissolved fraction reflecting the fact that the organic component of the particulate matter sampled is not the main carrier of cesium.**

The $^{137}$Cs/$^{134}$Cs activity concentration ratio in the dissolved phase changed drastically due to mixing with open-ocean water. In contrast, this ratio in the extracted organic fraction did not change with distance from shore or with $^{137}$Cs activity concentration and generally remained close to 1, even at locations far from the coast. This pattern indicates a decoupling between these two phases with the extracted organic fraction characterizing particle originating from land even at distance from the coast.

When considering $^{137}$Cs/$^{134}$Cs ratios, the source of radiocaesium in the coastal region north of FNPP1 should be related to releases from the core of Unit 1 of FNPP1, while in the coastal region south of FNPP1 the source should be a mixture of releases from the core of Unit 2 and the core of Unit 1 of FNPP1.

$^{137}$Cs activities on the Fukushima coast and the modified 7-day API showed a good positive relationship with the exception of 56N canal of FNPP1. In fact, at this location, high $^{137}$Cs activities are observed apart from an event of heavy rain.

This study shows the need to better characterize the inputs of rivers in taking into account on one side their hydrologic characteristics and on the other hand the nature of their solid discharges and phase change from particle to dissolved form. Special attention should be payed to the organic fraction of the riverine inputs as well as their fate in the coastal environment in the area which has been impacted by the accident.

-Data availability: it should be moved in the Material and methods?

A: As far as I understand, we need to have an independent section about Data availability based on Biogeoscience journal.

-References: please unify the format of periodicals. For example, Scientific Reports (Line 349); J. Radioanal. Nucl. Chem. (Line 356).......

A: Yes, we will do so.

-Figures: these figures are not clear, please redraw.... For example, fig.2 and fig.9.

A: Figures, 3 to 7 were revised and new figures are included in the reply to RC1. We will refine fig.2 and fig.9 later. We will delete figures 12-16 and 19 based on comments from RC1.

A: Technical issues stated below will be revised appropriately.
-Line 36: 'Nishihara et al. (Nishihara et al., 2012)' should be 'Nishihara et al. (2012)'.
-Line 43: 'Miura et al. (Miura et al., 2020)' should be 'Miura et al. (2020)'.
-Line 57: '(Nagao et al., 2014)' should be 'Nagao et al. (2014)'.
-Line 115: '...the Low Level Radioactivity Laboratory At some stations, ....' should be '...the Low Level Radioactivity Laboratory. At some stations, .....'.
-Lines 128, 320, 324: 'Dataset of 134Cs and 137Cs activity...' should be 'Dataset of $^{134}$Cs and $^{137}$Cs activity...'.
-Line 188: '.....in the rivers, indeed Naulier et al. (Naulier et al., 2017)' should be '......in the rivers. Indeed, Naulier et al. (2017)'.
-Line 224: 'Tsurutal et al., (Tsuruta et al., 2014)' should be 'Tsurutal et al. (2014)'.
-Line 227: '1.06 (+-10%)' should be '1.06 (±10%)'.
-Lines 235, 310: '0.92 (+-10%)' should be '0.92 (±10%)'.
-Lines 314-316: '7-day, 5-7day' should be '7-days, 5-7days'.
-Line 409, delete the Japanese language
-Lines 358-359, 410, 416, 420, 423, Superscript: $^{137}$Cs
-Table1, "+-" changes "±"

End of replies to RC2.

We also need to revise a part of method section as below.

**In the current text I made a copy paste mistake. We only used conc. H2O2 with pure water, and did not use conc. HNO3. The Actual method for getting the organic fraction is given below.**

**On the filters there were yellow-brown matters for all surface layer samples. Then we poured H2O2 step by step about 2 ml each time. We stopped adding H2O2 when the color disappeared.** The fraction **dealt with in this paper corresponds to this soft extraction.  We revised this part as appropriately as below.**

**In the current text, we stated that "Organic form of radiocaesium of the samples were obtained by disillusion of organic portion on the filter using concentrated nitric acid and concentrated hydrogen peroxide at all stations, then filled and dried up in a Teflon tube.".**

**We will revise as below based on our lab note by my staff.**

**We performed filtration using membrane filter with a pore size of 0.45-μm (Millipore HA). The filters were dried up and weighed to determine the mass of particles on the filter. Then, 4ml conc. H2O2 were poured on the filter placed in a plastic try three times with an interval of a few days, 5 Dec.2014, 8 Dec. 2014, 10 Dec. 2014 for all treated filters. And 4ml pure water were added on 8 Dec. 2014. Then if orange or brown parts still exist, we added pure water 4ml and 4ml con.H2O2 again on 15 Dec. 2014 to about half the number of filters. The end point was    the visual inspection of the color disappearance. The solutions were put in a Teflon tube and measured at Ogoya laboratory after drying. Therefore, in the revised article we will mention extracted organic fraction instead of organic particles**

End of reply.